# Cost-effectiveness analysis of a maternal pneumococcal vaccine in low-income, high-burden settings such as Sierra Leone

**Gizem M. Bilgin** *, **Syarifah Liza Munira**, **Kamalini Lokuge**, **Kathryn Glass**

National Centre for Epidemiology and Population Health, The Australian National University, Canberra, Australia

* gizem.bilgin@anu.edu.au

**Data Availability Statement:** We collated all data from publicly available data sources. All data and code can be viewed on our GitHub: https://github.

## Abstract

Maternal pneumococcal vaccines have been proposed as a method of protecting infants in the first few months of life. In this paper, we use results from a dynamic transmission model to assess the cost-effectiveness of a maternal pneumococcal polysaccharide vaccine from both healthcare and societal perspectives. We estimate the costs of delivering a maternal pneumococcal polysaccharide vaccine, the healthcare costs averted, and productivity losses avoided through the prevention of severe pneumococcal outcomes such as pneumonia and meningitis. Our model estimates that a maternal pneumococcal program would cost $606 (2020 USD, 95% prediction interval 437 to 779) from a healthcare perspective and $132 (95% prediction interval -1 to 265) from a societal perspective per DALY averted for one year of vaccine delivery. Hence, a maternal pneumococcal vaccine would be cost-effective from a societal perspective but not cost-effective from a healthcare perspective using Sierra Leone's GDP per capita of $527 as a cost-effectiveness threshold. Sensitivity analysis demonstrates how the choice to discount ongoing health benefits determines whether the maternal pneumococcal vaccine was deemed cost-effective from a healthcare perspective. Without discounting, the cost per DALY averted would be $292 (55% of Sierra Leone's GDP per capita) from a healthcare perspective. Further, the cost per DALY averted would be $142 (27% GDP per capita) from a healthcare perspective if PPV could be procured at the same cost relative to PCV in Sierra Leone as on the PAHO reference price list. Overall, our paper demonstrates that maternal pneumococcal vaccines have the potential to be cost-effective in low-income settings; however, the likelihood of low-income countries self-financing this intervention will depend on negotiations with vaccine providers on vaccine price. Vaccine price is the largest program cost driving the cost-effectiveness of a future maternal pneumococcal vaccine.

## 1. Introduction

The various presentations of pneumococcal disease, including pneumonia and meningitis, are a significant cause of morbidity and mortality in children worldwide [1]. The widespread

com/gizembilgin/SLE_maternalPneumococcal_model.

**Funding:** GMB was supported by the Australian Government Research Training Program during this study. No additional external funding was received for this study. The funder had no role in study design, data collection and analysis, decision to publish, or preparation of the manuscript.

**Competing interests:** The authors have declared that no competing interests exist.

adoption of pneumococcal conjugate vaccines (PCVs) has significantly reduced the burden of pneumococcal disease [2, 3]; however, the largest burden of pneumococcal disease is now in the first few months of life before childhood immunisations commence [4, 5].

Maternal vaccination with pneumococcal polysaccharide vaccines (PPVs) have been proposed as a strategy for protection in early childhood before PCV-derived immunity develops [6]. Maternal vaccines for influenza, tetanus and pertussis have already been demonstrated to provide protection to infants within their first few months of life [7–9]. To date, only preliminary randomised control trials and case-control studies of maternal PPV have been conducted [6]. These studies have observed no adverse outcomes of PPVs in pregnancy but have yielded inconclusive results regarding the efficacy of PPV as a maternal vaccine due to their small-scale. In a previous paper, we constructed a dynamic transmission model which demonstrated that a maternal pneumococcal vaccine could reduce incidence by 73% (range 49–88%) in children <1 month, and 55% (range 36–66%) in children 1–2 months old [10].

This paper explores whether a maternal PPV could be cost-effective in reducing the burden of infant pneumococcal disease. We focus on modelling a low-income setting with a well-established infant PCV schedule, since maternal pneumococcal vaccination is primarily proposed as a supplement to protect the youngest infants in these settings. We consider a year-round programme for maternal pneumococcal vaccination which, like maternal tetanus toxoid vaccination in low-resource settings, opportunistically vaccinates pregnant women who present to antenatal care.

## 2. Methods

### 2.1 Overview

In this study, we quantify the incremental cost-effectiveness of introducing a maternal PPV vaccine alongside an infant PCV schedule compared to continuing with infant PCV vaccination only. The impacts of a maternal pneumococcal vaccine on health outcomes have been quantified in a previous paper [10]; a summary of key characteristics of the dynamic transmission model is provided in the S1 Text. Here, we estimate the costs of introducing a maternal pneumococcal vaccine, and the healthcare costs averted through the novel use of this vaccine. We present results from both healthcare and societal perspectives for one hypothetical year of maternal vaccine delivery. The time horizon of the study was one year since maternally derived immunity is expected to wane after one year [11], and a maternal pneumococcal vaccine is anticipated to only affect the incidence of disease in infants, not cause wider or ongoing reductions in transmission [10]. We applied a discounting rate of 3% to all health benefits and costs in line with existing cost-effectiveness analysis in Sub-Saharan Africa [12–15]. Discounting was applied using a continuous time approach with uniform age weighting [16]. All prices were adjusted to 2020 United States Dollars (USD) using the International Monetary Fund's gross domestic product (GDP) deflators [17] following methods proposed by Turner and colleagues [18].

### 2.2 Study setting

Sierra Leone was the primary setting chosen for the model, given it is a low-income country with a very high burden of pneumococcal disease in early childhood. It is a West African nation in Sub-Saharan Africa, the region where half of all pneumococcal-associated deaths in children under five occur today [1]. Although PCV has been a routine vaccine in Sierra Leone since 2011 [19], there remains a high rate of infant mortality with 75 deaths per 1,000 live births [20]. The government of Sierra Leone are committed to reducing deaths in children under five, having introduced a Free Health Care Initiative in 2010 that waives all medical fees

for pregnant and breastfeeding women, and for children under 5 years of age [21]. As vaccination with PPV requires a single dose [22], we estimated maternal pneumococcal vaccine coverage from the 2019 Demographic Health Survey estimate that 97.4% of pregnant woman in Sierra Leone receive at least one dose of maternal tetanus vaccine [20].

## 2.3 Health outcomes

**2.3.1 Disease model.** The details of the disease model have been described in a previous paper [10]. In brief, we constructed a dynamic Susceptible-Infected-Suspectable (SIS) model. The model contained detailed age-structure for infants under two, with additional classes representing PCV-derived immunity. The model was fitted to the prevalence of pneumococcal-attributable acute respiratory illness. We introduced a maternal vaccine to this fitted model and used reductions in pneumococcal incidence to estimate reductions in pneumococcal-associated health outcomes. Our transmission model focused on examining the effects of a maternal vaccine on children under one year of age since maternally derived immunity is expected to wane after one year [23]. We did not consider indirect effects of a maternal vaccine on older age groups since previous modelling suggested that a maternal pneumococcal vaccine would not cause widespread or ongoing reduction in transmission in the community [10]. No adverse events related to the vaccine were included in the model since such events have not been documented in existing trial data [6].

**2.3.2 Translation of health outcomes to DALYs.** We translated estimates of health outcomes averted into disability-adjusted life years (DALYs) to align with previous cost-effectiveness analysis of pneumococcal vaccines [12–15, 24]. We determined years of life lost (YLL) using life expectancy at birth from United Nations population estimates for 2020–2024 [25]. We calculated years lived with disability (YLD) using disability weights from the Global Burden of Disease Study 2016 [26]. We employed the 'severe lower respiratory infection' weight (0.133; 95% CI 0.088–0.190) for all non-fatal invasive pneumococcal disease episodes, and 'moderate lower respiratory infection' weight (0.051; 95% CI 0.032–0.074) for all non-invasive pneumococcal disease episodes. The duration of symptomatic respiratory infection was assumed to be 10 days (9–12 days) [26]. We assumed that 24.7% of all pneumococcal meningitis cases developed lifelong meningitis sequalae [27] and applied the disability weight for 'severe motor plus cognitive impartments' (0.542; 95% 0.374–0.702), as per Ojal et al. [14], to these episodes. Point estimates for DALYs associated with pneumococcal outcomes are presented in S1 Table.

## 2.4 Vaccine program costs

We split vaccine program costs into two components: vaccine costs, and operational costs (Table 1). Vaccine costs included the costs to procure the vaccine and related injection

**Table 1. Vaccine program costing parameters with point estimates, distributions, and sources.** All prices are in 2020 USD.

| Parameter | Component description | Point estimate | Distribution | Reference |
|---|---|---|---|---|
| Cost of vaccine | Price per dose | $8.63 | Fixed | [28, 29] |
| | Wastage rate | 5% | Fixed | [19, 30] |
| | Freight costs | 4.5% of vaccine value | Fixed | [30] |
| Injection equipment | Bundled price for auto-disable (AD) syringes and safety boxes (per dose) | $0.044 | Fixed | [31] |
| | Wastage rate | 10% | Fixed | [30] |
| Operational costs | All non-vaccine and injections supply cost including the cost of personnel, training, transport, and social mobilisation (per dose) | $0.97 | Gamma (6.25,0.15) | [32] |

equipment. Operational costs encompassed supply chain costs and service delivery costs. We assumed that maternal pneumococcal vaccination would take place as part of routine antenatal care visits and therefore did not include direct non-medical costs such as patient travel time.

**2.4.1 Vaccine costs.**   Vaccine costs included the price, wastage and freight costs of the vaccine, syringes, and safety boxes. PPVs have not been widely utilised in low-income nations since the introduction of PCV. The WHO's 2020 Global Market Study for pneumococcal vaccines identified no low-income nation self-procuring PPV [33]. The study recognised that PPVs were primarily prescribed to adults over 65 and younger adults with comorbidities in high- and middle-income nations. Given the lack of a comparable setting, we estimated the price per dose of PPV from the Pan American Health Organisation's (PAHO) vaccine price list for 2020 [29]. PAHO's price per dose aligned with South Africa's public health system price per dose of PPV (adjusted to 2020) [28]. Estimates of vaccine wastage and freight costs were taken from existing data on PCV. PPV wastage is expected to be low due to the vaccine's presentation as a liquid single-dose vial.

Injection equipment costs were taken from United Nation Children's Fund (UNICEF) Supply Division prices, as recommended in Sierra Leones's Expanded Programme of Immunization Comprehensive Multi Year Plan (cMYP) [19, 31]. These costs aligned with previous in-country GAVI approved funding estimates for injection equipment [34]. UNICEF prices are listed as free carrier prices, meaning that they include freight costs.

**2.4.2 Operational costs.**   Operational costs included all non-vaccine and injection supply costs. That is, costs of transportation and storage, and costs of personnel to deliver the vaccine, program management and training. Operational costs for our hypothetical maternal pneumococcal vaccine were based on existing maternal tetanus vaccine data. The tetanus toxoid (TT) vaccine was the first maternal vaccine endorsed by the WHO and is the most widely used in low-income settings [35, 36]. Further, TT has similar temperature storage requirements to PPV [22, 37]. Sierra Leone's latest cMYP does not include maternal tetanus vaccine specific operational costs [19]. Instead, our estimate of operational costs was informed by a global review of cMYPs submitted to WHO and UNICEF [32]. Notably, this global review's estimates of operational costs for polio and measles aligned with the true operational costs for polio and measles reported in Sierra Leone [19, 32]. Further, the review's estimate for maternal vaccine operational costs aligned with the operational costs per dose of maternal tetanus vaccination in Liberia, a country of similar size and neighbour to Sierra Leone [38]. We modelled the distribution of operational costs using a gamma distribution, as per previous cost-effectiveness analysis [14].

## 2.5 Healthcare costs and productivity losses averted

Pneumococcal cases were divided between invasive pneumococcal disease (IPD) and all other pneumococcal-attributable acute respiratory illnesses (ARI) (Fig 1). IPD presentations included pneumococcal pneumonia, pneumococcal meningitis, and non-pneumonia non-meningitis (NPNM) IPD.

Cases were modelled as receiving outpatient care, inpatient care, and at-home care (Table 2). We estimated direct and indirect medical costs using a Ghanaian costing study of pneumococcal pneumonia and meningitis. Ghana and Sierra Leone are both Western African nations which lie within the pneumococcal meningitis belt and have an established PCV schedule using three primary doses without a booster (3p+0). The Ghanaian costing study calculated a broad range of direct medical costs including the cost of medication, diagnostic tests, and hospital staff salaries in addition to hospital bed days. Costs for NPNM IPD were taken from pneumococcal pneumonia costs, as per Ojal et al. [14].

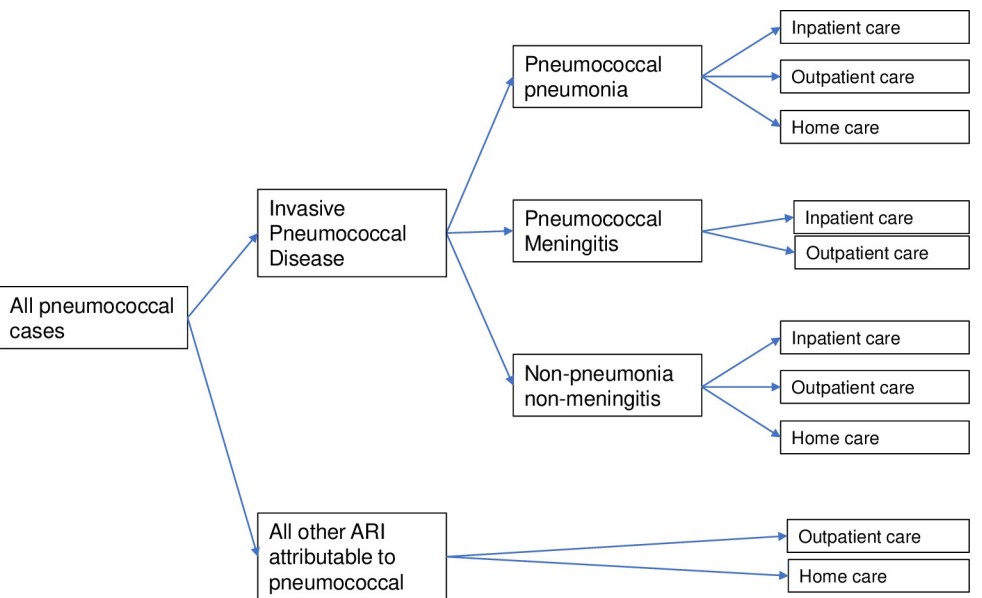

**Fig 1. Health outcome tree of pneumococcal presentations and types of care.**

We considered the loss of lifetime productivity due to premature death, and reduced productivity of caregivers while caring for infants using a human capital approach (Table 2) [39]. Lost productivity due to premature death was estimated using UN estimates of life expectancy

**Table 2. Proportion of cases with access to health care and health system costs per event of care with point estimates, distributions, and sources.** All costs presented in 2020 USD.

| Parameter | Presentation | Base estimate | Distribution | Reference |
|---|---|---|---|---|
| Access to care | Outpatient care | 85.7% | Beta(84,14) | [20] |
| | Home care | 14.3% | 1- Beta(84,14) | |
| | Inpatient care for pneumococcal pneumonia or NPNM | 65.2% | Beta(60,30) | [40] |
| | Inpatient care for pneumococcal meningitis | 76.1% | Derived from access to inpatient/outpatient care | [40] |
| Direct medical costs | Pneumococcal meningitis inpatient | $145.03 | Uniform(116,333)* | [40] |
| | Pneumococcal pneumonia or NPNM inpatient | $132.08 | Uniform(118,320)* | [40] |
| | Outpatient | $1.03 | Uniform(0.00,2.98) | [40] |
| | Home care | $1.34 | Uniform(0.00,2.98)* | [40] |
| Direct non-medical costs | Pneumococcal meningitis inpatient | $1.95 | Uniform(0.41,2.98)* | [40] |
| | Pneumococcal pneumonia or NPNM inpatient | $0.62 | Uniform(0.00,1.54)* | [40] |
| | Outpatient | $0.00 | Uniform(0.00,0.41)* | [40] |
| Caregiver productivity loss | Pneumococcal meningitis | $36.28 | Uniform(11.7,61.1)* | [40] |
| | Pneumococcal pneumonia | $13.05 | Uniform(3.29,48.8)* | [40] |
| Productivity loss due to premature death | Death | $13,282 (3% discounting) | NA | [25, 41] |
| | | $27,401 (0% discounting) | | |
| | | $9,202 (5% discounting) | | |

* Distribution as per correspondence with Ghanaian study author [40]

and World Bank estimates of gross national income per capita. Estimates for lost productivity of caregivers was taken from the Ghanaian study. We did not include productivity losses due to long-term sequalae.

Access to outpatient and inpatient care were modelled using care-seeking estimates from Sierra Leone's DHS and hospitalisation rates from the Ghanaian study. Patients who did not have access to outpatient care were assumed to receive home care. We assumed that all cases of pneumococcal meningitis require hospitalisation (inpatient care), however not all had access to hospital care [1, 40]. We assumed that non-invasive pneumococcal disease (all other ARI attributable to pneumococcal) did not require hospitalisation.

### 2.6 Sensitivity analysis

We conducted both probabilistic and one-way sensitivity analyses to explore the characteristics under which a maternal pneumococcal vaccine would be cost-effective in reducing infant pneumococcal disease. Probabilistic sensitivity analysis was conducted by Monte Carlo simulation, that is, by running the model 1000 times randomly drawing per-patient from the probability distributions of all underlying health outcome and costing parameters. The corresponding 1000 cost and 1000 health effect values were presented on a cost-effectiveness plane and summarised by a cost-effectiveness acceptability curve. We assumed that vaccine costs were fixed by government agreements and did not vary between individuals. Hence, we explored the effects of varied vaccine cost using one-way analysis.

### 3. Results

A maternal pneumococcal vaccine appears cost-effective from a societal perspective ($132 per DALY averted) but not from a healthcare perspective ($606 per DALY averted) using Sierra Leone's GDP ($527) as a threshold for cost-effectiveness (Table 3). Fig 2 presents the probability of a maternal pneumococcal vaccine being cost-effective with different cost-effectiveness thresholds and visualises uncertainty in expected incremental costs and DALYs averted.

The cost of delivering the vaccine program ($1,022,305 per 100,000 infants) had a greater influence compared to health system costs averted ($69,376 per 100,000 infants) over the expected incremental cost-effectiveness of the vaccine from a healthcare perspective. Most vaccine program costs (82.2%) were attributable to the cost of the vaccine (Table 4). Productivity loss due to premature mortality accounted for a majority ($745,005, 92%) of costs saved by a maternal pneumococcal vaccine from a societal perspective (S2 Table).

**Table 3. Cost per outcome averted under base model assumptions with 95% prediction interval from probabilistic sensitivity analysis.** All costs presented in 2020 USD.

| Outcome averted | Expected incremental cost | Expected incremental effect | Expected ICER of outcome | ICER 95% prediction interval |
|---|---|---|---|---|
| **Healthcare perspective** | | | | |
| DALY | 966,802 | 1,595 | 606 | 437 to 779 |
| Case | 996,802 | 948 | 1,017 | 937 to 1,096 |
| Hospitalisation | 966,802 | 258 | 3,740 | 2,710 to 4,779 |
| Death | 966,802 | 53 | 17,218 | 12,239 to 22,344 |
| **Societal perspective** | | | | |
| DALY | 211,455 | 1,595 | 132 | -1 to 265 |
| Case | 211,455 | 948 | 223 | 5 to 441 |
| Hospitalisation | 211,455 | 258 | 821 | -12 to 1,660 |
| Death | 211,455 | 53 | 3,764 | -46 to 7,549 |

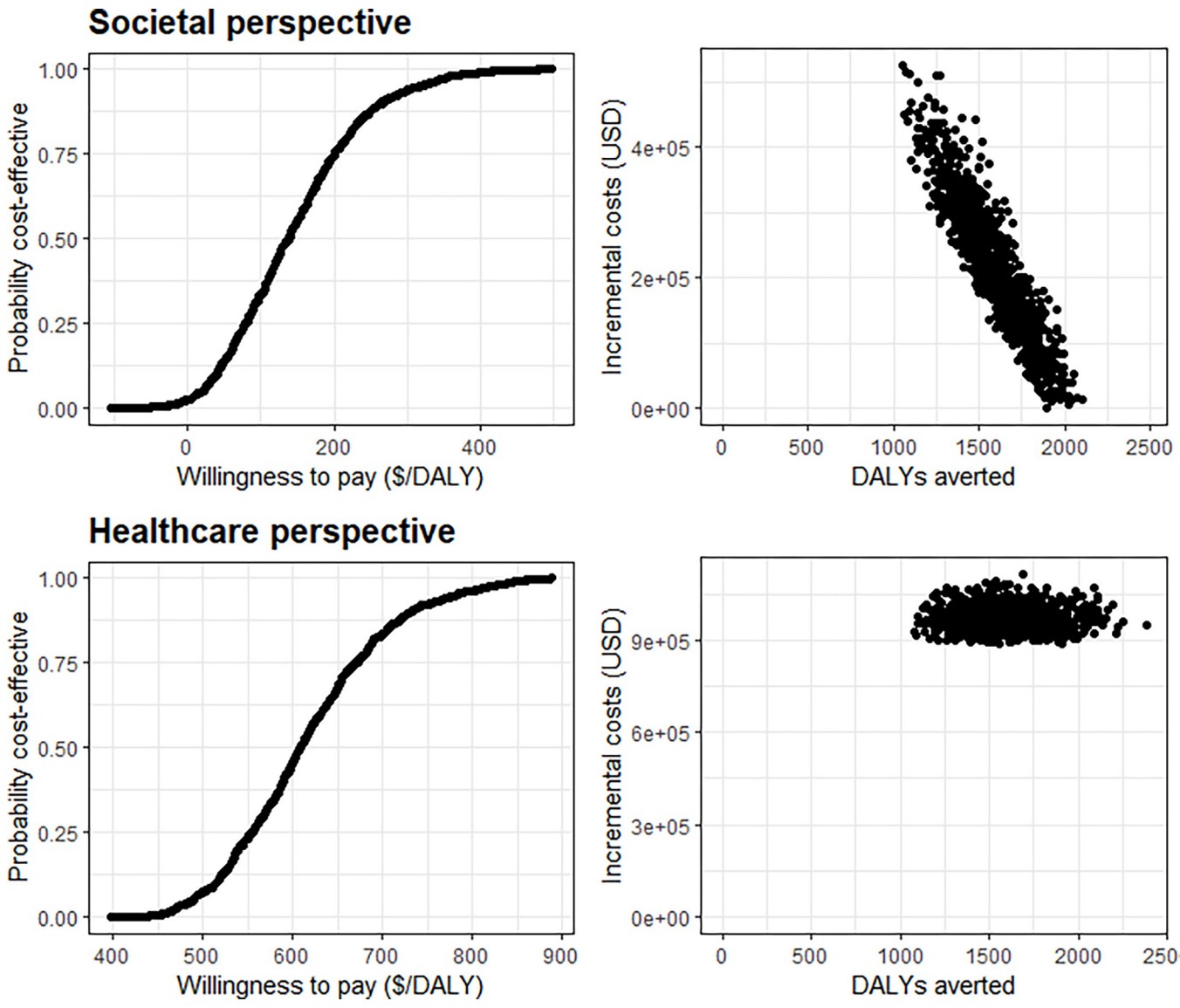

**Fig 2. Cost-effectiveness acceptability curve (left) and expected incremental costs and disability-adjusted life years (DALYs) averted (right) for a maternal pneumococcal vaccine in Sierra Leone.** Points represent 1,000 samples drawn with probabilistic sensitivity analysis.

**Table 4. Breakdown of maternal vaccine costs by program component.** All costs presented in 2020 USD.

| Component | Cost per 100,000 infants |
|---|---|
| Vaccine price | 840,821 (82.2%) |
| Vaccine dose wastage | 42,987 (4.2%) |
| Vaccine freight cost | 38,783 (3.8%) |
| Injection equipment (including wastage) | 4,716 (0.5%) |
| Operational costs | 94,998 (9.3%) |
| Total program cost | 1,022,305 (100.0%) |

## Healthcare perspective

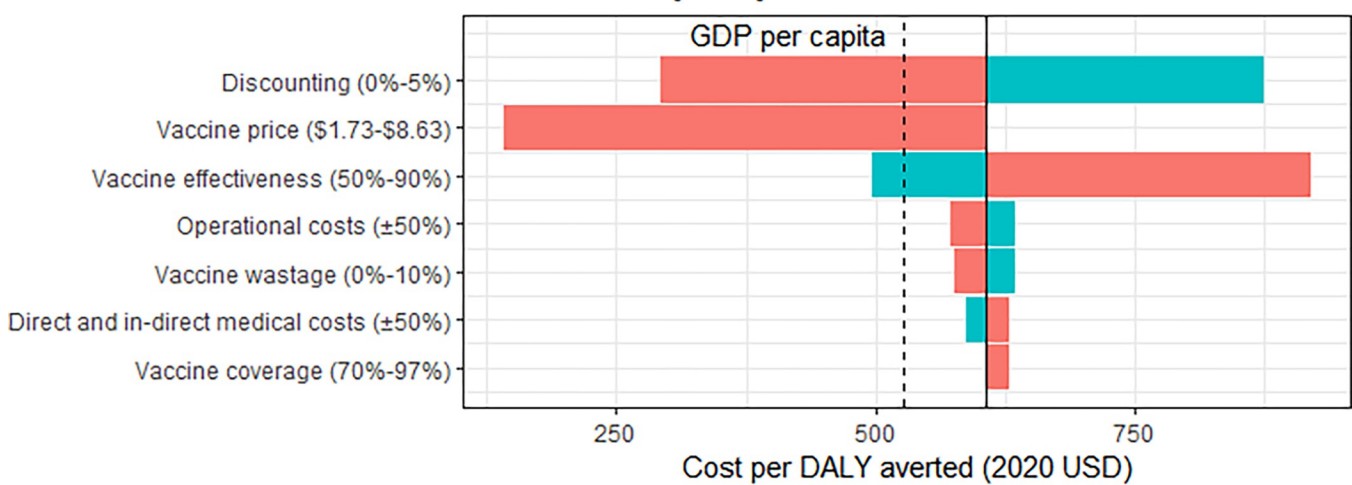

## Societal perspective

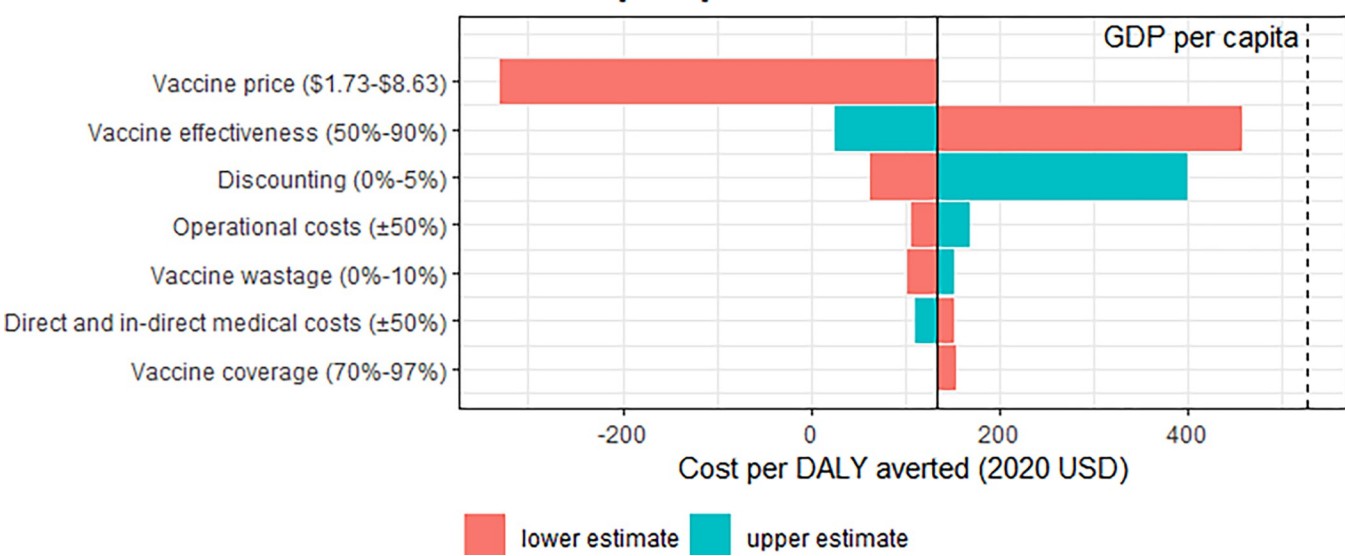

**Fig 3. Tornado diagram representing one-way sensitivity analysis.** The influence of key model parameters on incremental cost-effectiveness per DALY averted (2020 USD). Dashed line represents Sierra Leone's GDP per capita [17].

Sensitivity analysis identified health outcome discounting rates, vaccine price and vaccine effectiveness to be highly influential over cost per DALY averted (Fig 3 and S1–S3 Figs). Without discounting, the expected cost per DALY averted was cost-effective from both a healthcare perspective ($292, 55% of Sierra Leone's GDP) and a societal perspective ($66, 13% of Sierra Leone's GDP per capita). The vaccine would be cost-effective from a healthcare perspective ($142, 27% of Sierra Leone's GDP) and cost saving from a societal perspective if PPV could be procured at the same cost relative to PCV in Sierra Leone as on the PAHO reference price list. The vaccine would be cost-effective from a healthcare perspective under current market prices with an effectiveness of at least 85%.

## 4. Discussion

With current market prices, a maternal pneumococcal vaccine would be cost-effective from a societal perspective but not be cost-effective from a healthcare perspective in Sierra Leone using Sierra Leone's GDP per capita of $527 (2020 USD) as a cost-effectiveness threshold [17]. The choice of cost-effectiveness threshold and perspective should depend on the decision-maker's budget restrictions and local value judgements [42]. Our model identified vaccine price as the largest program cost driving the cost-effectiveness of a maternal PPV. This finding aligns with previous studies on the cost-effectiveness of new and underused vaccines, such as human papilloma virus (HPV), in low-income settings [43, 44]. Productivity losses due to pre-mature mortality were the main influence determining the difference between cost-effective-ness estimates from a societal and healthcare perspective, aligning with previous analysis of pneumococcal interventions in Ghana [40].

The implementation of a cost-effective maternal pneumococcal vaccine would require negotiations with vaccine suppliers on vaccine price. The cost of $10.22 to immunise a preg-nant mother with PPV is greater than the total cost $8.73/child of all existing childhood immu-nisation in Sierra Leone [45]. The PAHO PPV reference price is two thirds of PAHO's PCV reference price [29]. If a comparative PPV price were negotiated in Sierra Leone, a maternal pneumococcal vaccine would be cost-effective from a healthcare perspective, $142 per DALY averted (27% GDP per capita) and $2.09/child. Further, an effective maternal PPV could sig-nificantly increase the global demand for PPV, as only ~20 million doses are currently pur-chased annually [33]. Vaccine prices generally drop after introduction, especially through the increase in demand and involvement of humanitarian bodies such as GAVI [44].

Conventional methods of discounting the health effects of vaccines are being increasingly challenged [46–48]. In their critical review, Jit & Mibei establish that cost-effectiveness analyses of vaccines are particularly sensitive to discounting due to their distinctive characteristics com-pared to other health interventions [49]. Indeed, in this paper we demonstrate that the applica-tion of conventional 3% discounting shifts the cost-effectiveness of a maternal pneumococcal vaccine from $292 to $606 per DALY averted (from 55% to 115% of Sierra Leone's GDP per capita). Hence a decision maker's choice of discounting rates may decide whether a maternal pneumococcal vaccine is cost-effective.

A limitation of our model was that we did not include all non-invasive pneumococcal pre-sentations, such as otitis media and non-invasive pneumonia. Otitis media, in particular, causes a high burden of DALYs in Sub-Saharan Africa [50]. Additionally, we based hospitalisa-tion data on utilisation from DHS surveys. Given the limitations in access to hospitalisation in Sierra Leone, this is likely to under-estimate actual disease-related health care utilisation if access was adequate. Consequently, our model underestimates the health system costs averted by a maternal vaccine.

Operational costs for maternal tetanus toxoid (TT) vaccination may be an overestimation of future operational costs for a maternal pneumococcal vaccine. The tetanus toxoid vaccine is the only maternal vaccine currently included in Sierra Leone's Expanded Programme of Immunisation (EPI) [19]. We expect that the inclusion of an additional vaccine for an existing target group to be less costly. A recent systematic review identified that limited costing studies have been published detailing the costs of delivering maternal immunisation to pregnant women, particularly in low-income settings [36]. Future clinical trials should consider includ-ing a costing study component to better estimate the true operational costs of a maternal pneu-mococcal vaccine.

While our model provides a reasonable estimate of the cost-effectiveness of a routine mater-nal pneumococcal vaccine, our analysis does not include costs of introduction such as the

expansion of cold chain storage capacity, and updates to vaccination cards. We expect respective Ministries of Health and Finance to undertake more detailed planning specific to their setting before introduction. Botwright et al.'s systematic review estimates HPV introduction costs to be 46% of total financial costs, and 32% of economic costs in the first year of program delivery [51]. Notably, previous vaccine introduction costs in Sierra Leone have relied upon GAVI funding [34, 45, 52].

Maternal pneumococcal vaccines demonstrate the potential to be cost-effective in low-income settings; however, their introduction will require negotiations with a vaccine provider or funding support from a humanitarian body. The current market price for PPV is too high for a maternal pneumococcal program to be cost-effective from a healthcare perspective in our study setting of Sierra Leone. Further, the future use of a maternal PPV would require advanced planning with suppliers to avoid supply shortages.

## Supporting information

**S1 Text. Dynamic transmission model summary.**
(DOCX)

**S1 Table. DALYs associated with different outcomes of pneumococcal infection.**
(XLSX)

**S2 Table. Breakdown of expected costs and health outcomes with and without a maternal pneumococcal vaccine.** Outcomes presented for as the mean result over 1,000 simulations for 100,000 children.
(XLSX)

**S1 Fig. Sensitivity analysis of discounting rate.** Cost-effectiveness acceptability curve (left) and expected incremental costs and disability-adjusted life years (DALYs) averted (right) for a maternal pneumococcal vaccine in Sierra Leone using varying discounting rates on costs and health outcomes. Points represent 1,000 samples drawn with probabilistic sensitivity analysis.
(TIFF)

**S2 Fig. Sensitivity analysis of vaccine price.** Cost-effectiveness acceptability curve (left) and expected incremental costs and disability-adjusted life years (DALYs) averted (right) for a maternal pneumococcal vaccine in Sierra Leone using varying vaccine price estimates. Points represent 1,000 samples drawn with probabilistic sensitivity analysis.
(TIFF)

**S3 Fig. Sensitivity analysis of maternal vaccine effectiveness.** Cost-effectiveness acceptability curve (left) and expected incremental costs and disability-adjusted life years (DALYs) averted (right) for a maternal pneumococcal vaccine in Sierra Leone varying the effectiveness of the maternal vaccine in preventing severe outcomes in infants. Points represent 1,000 samples drawn with probabilistic sensitivity analysis.
(TIFF)

## Acknowledgments

We acknowledge that this study would not have been possible without the efforts of previous studies used to inform our costing.

## Author Contributions

**Conceptualization:** Gizem M. Bilgin, Syarifah Liza Munira, Kamalini Lokuge, Kathryn Glass.

**Formal analysis:** Gizem M. Bilgin.

**Methodology:** Gizem M. Bilgin, Syarifah Liza Munira, Kathryn Glass.

**Software:** Gizem M. Bilgin.

**Supervision:** Syarifah Liza Munira, Kamalini Lokuge, Kathryn Glass.

**Validation:** Kamalini Lokuge.

**Visualization:** Gizem M. Bilgin.

**Writing – original draft:** Gizem M. Bilgin.

**Writing – review & editing:** Gizem M. Bilgin, Syarifah Liza Munira, Kamalini Lokuge, Kathryn Glass.

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
