## [Decision Letter · Decision Letter 0]

12 May 2023

PGPH-D-22-01212

Cost-effectiveness analysis of a maternal pneumococcal vaccine in low-income, high-burden settings such as Sierra Leone

Dear Gizem Bilgin,

Thank you for submitting your manuscript to PLOS Global Public Health. After careful consideration, we feel that the manuscript does not fully meet PLOS Global Public Health’s publication criteria as it currently stands. The reviewers have carefully reviewed the manuscript and provided detailed feedback. We invite you to submit a revised version of the manuscript that appropriately addresses the points raised during the review process.

We look forward to receiving your revised manuscript.

Kind regards,

Edina Amponsah-Dacosta, Ph.D., MPH

Academic Editor

Journal Requirements:

Additional Editor Comments (if provided):

Reviewers' comments:

Reviewer's Responses to Questions

**Comments to the Author**

1. Does this manuscript meet PLOS Global Public Health’s publication criteria? Is the manuscript technically sound, and do the data support the conclusions? The manuscript must describe methodologically and ethically rigorous research with conclusions that are appropriately drawn based on the data presented.

Reviewer #1: Partly

Reviewer #2: Yes

Reviewer #3: Yes

2. Has the statistical analysis been performed appropriately and rigorously?

Reviewer #1: N/A

Reviewer #2: Yes

Reviewer #3: N/A

3. Have the authors made all data underlying the findings in their manuscript fully available (please refer to the Data Availability Statement at the start of the manuscript PDF file)?

Reviewer #1: No

Reviewer #2: No

Reviewer #3: Yes

4. Is the manuscript presented in an intelligible fashion and written in standard English?

Reviewer #1: Yes

Reviewer #2: Yes

Reviewer #3: Yes

5. Review Comments to the Author

Reviewer #1: This study aimed to examine the cost-effectiveness of pneumococcal vaccines among pregnant women. Although the topic is of importance, there are a few concerns

1. The major concern is that the study was performed merely from the health system perspective. The second panel on cost-effectiveness in health and medicine has recommended that the cost-effectiveness analysis be conducted from both societal perspective and health perspective.

2. Another major concern is the reporting does not confirm CHEER good practice. Please check it. Some key parameters are missing, which does not allow other others to repeat the analysis. For example,

a. How will the vaccine be vaccinated among pregnant women?

b. What are efficacy rates applied to different age of the children?

c. What are the incidence rates of pneumococcal-associated illness among children under 1 month, and children between 1-2 months,

d. What is the share of conditions under different treatment settings and their associated health outcomes (e.g., death rate)

3. Regarding the cost, health care cost is higher than outpatient cost, which seems not reasonable. Also, the cost of outpatient costs less than 1 dollar is not reasonable either. Please note that WHO-CHOICE data do not include the cost of medicine and lab tests. It is not clear whether those costs are converted to 2020 USD.

4. It would be better to report outcomes under two scenarios (no vaccine and with vaccine), which could help readers understand better whether the estimation is reasonable, instead of ICER directly.

5. The 95% CI of ICER is quite narrow. Given there are so many parameters included in the analysis, the accuracy of the estimation is skeptical.

6. The conclusion is worth debating even with the results of $570/DALYs. There are different cut-offs have been proposed, including one published in Lancet Global Health that uses 1.5 GDP/capita as cut-off. The choice of threshold should be discussed.

Reviewer #2: This study uses a previously published transmission model to determine the cost-effectiveness of implementing a maternal pneumococcal vaccine in Sierra Leone. The paper is well written. The novelty of this study is in the parameterisation cost-effectiveness model; the impact of maternal vaccination has been previously published. Though the parameterisation of the model is well-thought-out, the outcomes of the cost-effectiveness analysis are lacking, giving a disappointing amount of information, and limiting its useability to inform policymakers. Further work on the cost-effectiveness analysis will greatly improve the paper. I’ve made some recommendations on how to improve these results below. In addition, the code which produced the study should be made freely available so that others can check the author’s work for reproducibility. This work should not be in a peer-reviewed journal if the code cannot be peer-reviewed. If these points are addressed this paper should be published.

Major points

What are the incremental costs and incremental health benefits of implementing the maternal programme? This needs to be explicitly stated front and centre. Further, a cost-effectiveness acceptability curve/frontier would be a great assessment of this study so that the readers can see how the cost-effectiveness changes given a varied wtp threshold. The standard of using GDP per capita as a fixed measure of the cost-effectiveness threshold is dated and having this information in the form of a curve with varying wtp thresholds would be much more informative for decision-makers. In addition, a plot showing the uncertainty is crucial, something like a CEA plane using ICERs or Incremental Net Monetary Benefit or a plot of the EPVI/EPPVI given a wtp threshold would greatly enrich the study. A couple of papers which the authors might find useful are (note neither of these papers is affiliated with the reviewer):

https://pubmed.ncbi.nlm.nih.gov/31104743/

https://journals.sagepub.com/doi/full/10.1177/0272989X211045070

Though probably mentioned in their model paper, there needs to be some further information in this paper on how the maternal vaccine is implemented in a dynamic model for completeness. E.g. does the model take into account the protection against pneumococcus in mothers themselves? If so, how? Does the model consider the potential reduction in pneumococcal transmission between mother and infant? What are the indirect effects in total like? A compartmental figure showing this in an Appendix would be very useful. How was the waning introduced? What is the waning function in the dynamic model (this is a crucial component) it says 1 year but is this lost exponentially?

What is the time horizon for this study? It appears to be one year? Why? Depending on how the waning of immunity is implemented, there could be multi-seasonal effects no? A much longer time horizon needs to be considered, at least 10 years, unless the authors have a good reason why(?) Further, the discounting needs to be applied to health benefits and costs. The conclusion that the choice of discounting rate is significant in determining cost-effectiveness is flawed if only implied to the DALYs, it implies that the choice of DALYs is significant in determining cost-effectiveness.

Is the code accessible in any way? Neither this paper nor the modelling paper it references has its code online to share to check for reproducibility. Modelling papers, including cost-effectiveness analysis, should have their code freely available. This work should not be in a peer-reviewed journal if the code cannot be peer-reviewed.

Minor points

The only programme considered appears to be a year-round maternal programme (not sure this is made explicit in the methods)? Would a seasonal programme be feasible in Sierra Leone (I’m not familiar with the epi there sorry!)? Perhaps a brief overview of the epi in the introduction would help contextualise the work within the global landscape.

There appears to be very little uncertainty in the model looking at Table 3, could the authors comment on this? I.e. how good is the inherent uncertainty in the model in capturing the true uncertainty? Again, perhaps creating the plots described above might help answer this point.

Tables

Could the DALY estimates be added to a table with uncertainty (maybe 1 or 2)?

Figures

Figure 1.

Could point-estimate proportions be added to the lines for Figure 1? This would help give a diagrammatic overview of the cost-effectiveness model.

Figure 2.

Add grid lines to the plot.

Remove the underscores from the legend.

Label the dashed line.

Reviewer #3: This article estimates the cost-effectiveness of implementing pneumococcal maternal vaccination in Sierra Leone. The health-economic model is plugged on top of a previously published dynamic transmission model. The article is clear, concise, and well written.

I would suggest the following minor improvements:

1) In the abstract: “95% prediction interval” rather than “range”.

2) In the abstract: for the results you are mentioning, specify “over one year of vaccine delivery” or “for one year of vaccine delivery” since the benefit of vaccine protection is lifelong and thus not limited to the hypothetical year of implementation.

3) In the abstract: “the choice of discounting rates for health outcomes determines” => specify that this is discounting “over individuals’ life” (maybe find a better way to say this). Just to avoid any confusion with “discounting over the period of vaccine delivery”, which is something different and not what you are doing in the paper as I understand it.

4) I would mention the dynamic transmission model earlier in the paper (e.g. in the Introduction or the abstract). Otherwise some readers might discard the paper too fast, thinking it is just using a static model.

5) Line 72. Remove “, so” ?

6) Line 87. “There is high uptake of a maternal tetanus vaccine in Sierra Leone, with 97.4% of pregnant women receiving at least one dose during pregnancy [14]. A single dose of PPV is sufficient [16].” => This part sounds a bit strange. Do you mean that you will assume a 97.4% coverage for PPV based on the maternal tetanus vaccine coverage, and that PPV is a single dose vaccine?

7) Line 97. “For children under 1”. Are indirect effects on older age groups negligible? Is this something you found in your previous study?

8) Line 108. First use of “IPD” => define it here.

6. PLOS authors have the option to publish the peer review history of their article (what does this mean?). If published, this will include your full peer review and any attached files.

**Do you want your identity to be public for this peer review?** For information about this choice, including consent withdrawal, please see our Privacy Policy.

Reviewer #1: No

Reviewer #2: No

Reviewer #3: No

---

## [Decision Letter · Decision Letter 1]

3 Jul 2023

Cost-effectiveness analysis of a maternal pneumococcal vaccine in low-income, high-burden settings such as Sierra Leone

PGPH-D-22-01212R1

Dear Gizem Bilgin,

We are pleased to inform you that your manuscript 'Cost-effectiveness analysis of a maternal pneumococcal vaccine in low-income, high-burden settings such as Sierra Leone' has been provisionally accepted for publication in PLOS Global Public Health.

Best regards,

Edina Amponsah-Dacosta, Ph.D., MPH

Academic Editor

Reviewer Comments (if any, and for reference):

Reviewer's Responses to Questions

**Comments to the Author**

1. If the authors have adequately addressed your comments raised in a previous round of review and you feel that this manuscript is now acceptable for publication, you may indicate that here to bypass the “Comments to the Author” section, enter your conflict of interest statement in the “Confidential to Editor” section, and submit your "Accept" recommendation.

Reviewer #2: All comments have been addressed

Reviewer #3: All comments have been addressed

2. Does this manuscript meet PLOS Global Public Health’s publication criteria? Is the manuscript technically sound, and do the data support the conclusions? The manuscript must describe methodologically and ethically rigorous research with conclusions that are appropriately drawn based on the data presented.

Reviewer #2: Yes

Reviewer #3: Yes

3. Has the statistical analysis been performed appropriately and rigorously?

Reviewer #2: Yes

Reviewer #3: Yes

4. Have the authors made all data underlying the findings in their manuscript fully available (please refer to the Data Availability Statement at the start of the manuscript PDF file)?

Reviewer #2: Yes

Reviewer #3: Yes

5. Is the manuscript presented in an intelligible fashion and written in standard English?

Reviewer #2: Yes

Reviewer #3: Yes

6. Review Comments to the Author

Reviewer #2: The authors has addressed my comments and made appropriate changes to the study. In this format the paper is ready for publication.

Reviewer #3: My comments have been addressed

7. PLOS authors have the option to publish the peer review history of their article (what does this mean?). If published, this will include your full peer review and any attached files.

**Do you want your identity to be public for this peer review?** For information about this choice, including consent withdrawal, please see our Privacy Policy.

Reviewer #2: No

Reviewer #3: No
